# Keeping in step with the young: Chronometric and kinematic data show intact procedural locomotor sequence learning in older adults

**Leif Johannsen**[1,2]*, **Erik Friedgen**[1], **Denise Nadine Stephan**[1], **Joao Batista**[2], **Doreen Schulze**[2], **Thea Laurentius**[2], **Iring Koch**[1], **Leo Cornelius Bollheimer**[2]

**1** Institute of Psychology, RWTH Aachen University, Aachen, Germany, **2** Medical Clinic VI–Geriatrics, University Hospital RWTH Aachen University, Aachen, Germany

* Leif.Johannsen@psych.rwth-aachen.de

## Abstract

Sequence learning in serial reaction time tasks (SRTT) is an established, lab-based experimental paradigm to study acquisition and transfer of skill based on the detection of predictable stimulus and motor response sequences. Sequence learning has been mainly studied in key presses using visual target stimuli and is demonstrated by better performance in predictable sequences than in random sequences. In this study, we investigated sequence learning in the context of more complex locomotor responses. To this end, we developed a novel goal-directed stepping SRTT with auditory target stimuli in order to subsequently assess the effect of aging on sequence learning in this task, expecting that age-related performance reductions in postural control might disturb the acquisition of the sequence. We used pressure-sensitive floor mats to characterise performance across ten blocks of trials. In Experiment 1, 22 young adults demonstrated successful acquisition of the sequence in terms of the time to step on the target mat and percent error and thus validated our new paradigm. In Experiment 2, in order to contrast performance improvements in the stepping SRTT between 27 young and 22 old adults, motion capture of the feet was combined with the floor mat system to delineate individual movement phases during stepping onto a target mat. The latencies of several postural events as well as other movement parameters of a step were assessed. We observed significant learning effects in the latency of step initiation, the time to step on the target mat, and motion parameters such as stepping amplitude and peak stepping velocity, as well as in percent error. The data showed general age-related slowing but no significant performance differences in procedural locomotor sequence learning between young and old adults. The older adults also had comparable conscious representations of the sequence of stimuli as the young adults. We conclude that sequence learning occurred in this locomotor learning task that is much more complex than typical finger-tapping sequence learning tasks, and that healthy older adults showed similar learning effects compared to young adults, suggesting intact locomotor sequence learning capabilities despite general slowing and normal age-related decline in sensorimotor function.

**Data Availability Statement:** The aggregated data files for this publication can be accessed under the

following DOI: https://doi.org/10.6084/m9.figshare.19487957.v1.

**Funding:** Funded by the Federal Ministry of Education and Research (BMBF) and the Ministry of Culture and Science of the German State of North Rhine-Westphalia (MKW) under the Excellence Strategy of the Federal Government and the Länder ((DE-82) EXS-SF-OPSF514). In addition, this study was supported by the Robert Bosch Stiftung in the context of the "Lehrstuhl für Geriatrie an der Medizinischen Fakultät der RWTH Aachen" (32.5.1140.0009.0). The funders had no role in study design, data collection and analysis, decision to publish, or preparation of the manuscript.

**Competing interests:** The authors have declared that no competing interests exist.

## Introduction

The learning of structured sequences of keypresses as in the serial-reaction-time-task (SRTT) paradigm [1] is an established, lab-based paradigm to investigate the acquisition of motor and cognitive skills. The SRTT paradigm is one of the best approaches to illustrate procedural and implicit knowledge formed during the acquisition of a sensorimotor skill. In this experimental paradigm, individual manual key presses are required in response to a seemingly random sequence of visual or auditory cues. For example, four adjacent reaction buttons are spatially assigned to four stimulus positions on the computer screen. Performance with seemingly irregular but with actually predictable stimulus sequences is contrasted against performance with randomized, unpredictable series of stimulus sequences. Exposure to such sequences across several blocks results in a performance curve, for example in terms of decreasing response latency, demonstrating the gradually increasing ability to predict the upcoming sequence. Nissen and Bullemer [1] demonstrated that the production of the motor sequence typically becomes more automatic as learning progresses. In order to separate general practice-related performance improvements and sequence-specific learning, a block with purely randomized stimuli is often inserted in the experiment to uncover differences in performance between late blocks with a fixed sequence and the random block ["negative transfer"; see, e.g., 2–5 for reviews]. Any regularity contained within the stimulus sequence will result in improvements in response latencies and error rates beyond a performance level achieved by just practicing the response task itself. Instead, participants acquire the ability to predict upcoming stimuli and to anticipate any subsequent responses [1,6].

An important aspect in sequence learning of the SRTT is the question of whether participants require conscious awareness of the regularities that are embedded in a sequence [7]. Interestingly, participants are not necessarily able to report any hidden regularity as they learned the sequence without being aware of it [8] and the ability to report a sequence explicitly is not necessary for the learning of a sequence to occur [2,4,9, for reviews]. It has been demonstrated, however, that being consciously aware of the sequence facilitates its acquisition [e.g. 10]. In addition, multimodal contingent consequences of any responses during learning enhance acquisition of explicit knowledge about a stimulus sequence [11–13]. Haider and coworkers interpreted this enhancement by way of the ideomotor principle of action control as well as an increased sense of agency, which might trigger a causal inference process [11,13].

However, it is notable that the typically used motor task in sequence learning consisted of relatively simple actions such as finger key presses, which do not provide a rich sensorimotor context as observed in gross movements such as goal-directed upper-limb reaching or stepping with the limbs. As finger key presses are actions of relative short duration, they do not encompass the same amount of feedback-driven control during movement execution as in movements of body segments or the full body with much longer movement durations. Gross movement tasks on the other hand provide an abundance of movement degrees of freedom and thus multitude of possible solutions. Therefore, practice effects may express themselves on several potential parameters, e.g. spatiotemporal. By measuring not only the onset latencies but also movement times and the motion kinematics during sequence learning, however, it may be possible to better distinguish improved motor performance due the acquisition of a sequence from the optimisation of movements due to practice [14,15].

Our central goal of the present study consisting of two experiments was to evaluate the effects of older age on performance in a locomotor SRTT task, which required full body movements. The first experiment validated a novel locomotor sequence learning task. In the second experiment, using this novel, more complex locomotor task, we then assessed whether there are age-related differences in sequence acquisition. In locomotor sequences, the motor system

must ensure that the body balance remains stable when a person takes one or more steps. The adaptation of stepping behaviour to dynamic environmental stimuli, especially if these occur in the form of sensory or mechanical perturbations that could destabilize body balance, is an important sensorimotor capability to avoid physical harm. The detection of regularities in environmental dynamics allows the preparation of postural adjustments to minimize the impact of any balance disturbance. This learning to predict the sequence of environmental stimuli is important to prevent an unintended destabilization of balance and a fall [16]. The complexity of postural and locomotor responses stems from the planning of sequences of several postural adjustments to ensure postural stability before any voluntary response can be executed. Anticipation and prediction play important roles in keeping body balance stable during standing or walking as disturbances of body balance, either self-imposed or externally caused, can be derived from context-specific knowledge gained through experience.

Sequence learning in a more complex locomotor task such as stepping has not been investigated in detail until now but experimental paradigms in terms of either postural or locomotor sequence learning have been developed in recent years. These studies indicated that sequence learning is likely to occur in more complex movement contexts. It seems, however, that the biomechanical constraints imposed on the body during full body movements, such as keeping body balance stable during stepping, may interfere with those processes of sequence learning as they are normally observed in manual SRTTs. For example, Du et al. [17] utilized a lower-limb foot-tapping SRTT, which more closely resembled the original manual SRTT. Du and Clark [18] concluded that the implicit sequence knowledge arose from statistical learning during foot-tapping sequence learning. This conclusion was based on the observation that responses to stimuli were only correlated with the single preceding stimulus or response, so that participants learned the probabilities by which a certain stimulus predicted a subsequent stimulus. Indications of chunking were also observed in the foot-tapping SRTT but were interpreted as masquerading biomechanical task constraints, such as improved strategies to control body weight shifts [18].

The study of locomotor sequence learning is important for understanding how older adults deals with such complex motor tasks. Older adults show an increasing fall risk [19]. Locomotor performance in older adults can inter-individually be restricted by many factors, such as reduced muscle strength and impaired cognitive and sensorimotor functions, which may lead to reductions in preferred gait speed, step and stride length, cadence, duration of the swing and single stance phases, and generally decreased joint range of motion, but also increases in postural stiffness, greater attentional demands, and stronger visual dependency [20]. Older adults with increased fall risk show longer stepping latencies in a choice stepping reaction time test by elongating anticipatory postural adjustments, thereby prioritizing dynamic stability over response time [21]. This observation corresponds to the "posture-first" principle commonly observed in older adults when confronted with an additional cognitive load in multitasking situations [i.e. 22]. Reduced APA amplitude and increased duration or absent APA components correlate with an increased fall risk in older adults [23,24]. In addition, Cohen et al. [25] demonstrated that step initiation in a reaction time task is delayed in older adults' step initiations, possibly due to less efficient inhibition of erroneously response tendencies [26–28].

The effect of ageing does not influence motor skill acquisition in general but rather in a manner specific to the characteristics of the motor task to be learnt [29]. This applies to sequence learning too. Urry et al. [30] reported preserved SRTT performance in terms of manual response latency gains during sequence learning in older adults. Similarly in manual tasks, Bhakuni and Mutha [31] found no learning reductions in older adults, while Verneau et al. [32] observed ageing-related reductions for explicit learning of a sequence but not for implicit learning. In more general terms, no ageing effects were found for learning in SRTTs with

relatively simple sequences [e.g. 33,34–37]. With more structurally complex sequences, however, learning in a SRTT declines with older age [e.g. 38,39–41]. Increases in complexity may be expressed not only in the sequences but also in the types of movements, so that when the required movements or actions become more complex, ageing effects become more likely too. For example, Hayes et al. [42] showed that older adults need more time to learn a sequence in a postural tracking task.

By moving away from a more cognitive focus on motor sequence learning involving simple manual button presses, we evaluated sequence learning in a locomotor activity as a function of ageing by recording performance in locomotor sequence learning. This included movement parameters such as latencies of step initiation and stepping on the target, but also step lengths, step velocities, and movement times in addition to traditional parameters (response latencies and error rates). Based on the extant evidence, we expected that participants would acquire knowledge of the regularities in the sequences and optimize their movement production in advance. This should be visible in performance curves and learning effects by means of reduced latencies and error rates. In addition, we expected that the performance limitations observed and the prioritization of postural stability during walking would limit the ability to acquire knowledge about specific locomotor sequences in older adults. According to this hypothesis, older adults should demonstrate a reduced ability to acquire a complex sequence of stimuli and motor responses. On the other hand, locomotor activities provide rich and meaningful state-dependent and contextual, multimodal sensorimotor cues, by which movement sequences could be discretized into specific sensorimotor events. This could facilitate the extraction of regularities in complex but predictable sequences. For example, reversals in movement direction may represent distinct sensorial signals indicative of regular sensorimotor state alterations useful for sequence segmentation and which older adults might benefit from.

## Experiment 1

This experiment was planned to establish the novel methodology for the investigation of a locomotor SRTT demonstrating the basic locomotor sequence learning effects, and enable us to calculate an adequate statistical power for Experiment 2. Further, observation of participants' behaviour allowed us consider suitable performance parameters for the evaluations of age-related changes in locomotor sequence learning.

## Methods

### Participants

The age inclusion criterion was an age between 18 and 35 years. We recruited 22 healthy younger adults (18 females and 4 males; average age = 22.0 +/- 3.3 years) from the RWTH Aachen University. The research ethics review board of the medical faculty granted ethical approval for this study (EK 322/19). All participants provided written informed consent before inclusion in the study according to the Declaration of Helsinki.

### Stimuli and task

The administration of the experimental stimuli was programmed in PsychoPy2 [43]. The target stimuli 1, 2, 3, and 4 (1 and 3 were target fields for the left foot; 2 and 4 for the right foot) were presented auditorily as verbal number heard stimuli spoken by a male voice via stereo headphones ("eins", "zwei", "drei", "vier" in German; JVC stereo headphones HA-S31M) and presented with a stimulus duration of approximately 470 ms and at a sound pressure level of about 65 dB. Three different types of steps were possible as a single response to an auditory

stimulus: a forward step, a backward step, a step with the same foot on the spot. A step on the spot did not occur as a direct stimulus repetition but could be the result of an interspersed step of the other foot, for example such as a right-left-right triple (2-3-2). In response to the presentation of a single stimulus, participants were instructed to place their foot onto one of four numbered target fields (pressure-sensitive mats) as fast as possible following stimulus presentation. The accuracy of stepping onto the correct target was emphasized, while it was declared unimportant, whether or not a target was hit with the full foot.

In contrast to a manual SRTT, where response keys are typically released immediately after having been pressed, this locomotor SRTT did not require participants to return to a default, neutral placement of the feet after each single step. Instead, they were instructed to remain in the current standing position. The maximum response latency for a trial was 3 s with an inter-trial-interval following a response of 1 s. Responses had to be performed within the normal trial period or a trial was registered as incorrect. Following an incorrect stepping response, an "oops" sound was played as error feedback. The error feedback duration was 400 ms with an additional 100 ms of silence. The target of an actual step had to be coded "online" manually by the experimenter during the response-stimulus interval. This was necessary due to the high sensitivity of the pressure mats, which would detect weight shifts onto the standing leg also and would register these events as steps by standing leg additionally to the actual step.

The four numerical stimuli were arranged in sequences of 12 numbers so that two different second-order conditional sequences [44] were employed counterbalanced across participants (sequence 1: 2-4-1-2-3-2-1-4-3-1-3-4, sequence 2: 2-3-4-1-2-4-2-1-3-1-4-3). These sequences had been used in previous experiments [10,45]. Immediate stimulus repetitions were excluded in both sequences and both were comparable regarding the number of reversal patterns with each sequence (e.g. 2-3-2, 3-1-3, 2-4-2, 1-3-1). Randomized sequences were subject to the following constraints. Each number occurred with the same frequency. Further, randomized sequences did not contain any repetitions of identical elements (e.g. 2–2). Ascending or descending full runs (e.g. 1-2-3-4 or 4-3-2-1) and partial runs (e.g. 1-2-3-2-1 or 4-3-2-3-4) were possible.

Each sequence was presented 6 times per block, so that for a single block 72 individual trials were tested. In total 10 blocks were assessed. While the sequence followed a fixed pattern of the respective assigned sequence in the Blocks 3 to 8 and the 10th block, randomized stimulus sequences of digits were presented in the first, second and 9th block. For these three blocks, a randomized sequence was generated each, so that every participant received a unique set of three randomized blocks. Participants performed a practice block of 12 random stimuli at the start of the experiment to familiarize themselves with the experimental procedure. After data collection had been completed, a structured follow-up interview was conducted with each participant to ask whether they had detected any regularities on the sequence of target stimuli or whether they could even recall parts of or their entire sequence explicitly. The questioning began with asking for anything remarkable about the experiment and specific details, if a participant had to report an observation. In the next step, participants were asked whether they had detected any regularities and for further details if they affirmed this. If a participant mentioned the impression of a sequence of targets or steps, then the participant was asked to recall any elements of the sequence. A reported partial sequence was only scored as accurate, if it consisted of at least 4 elements in their correct order.

## Experimental setup and procedure

Four target fields (pressure-sensitive mats; 23 x 16 cm) were placed on the lab floor arranged in a 2 x 2 squared configuration (distances between mats: 27 cm in anteroposterior direction, 18 cm in mediolateral direction; stepping area: 73 x 50 cm). Participants were instructed to

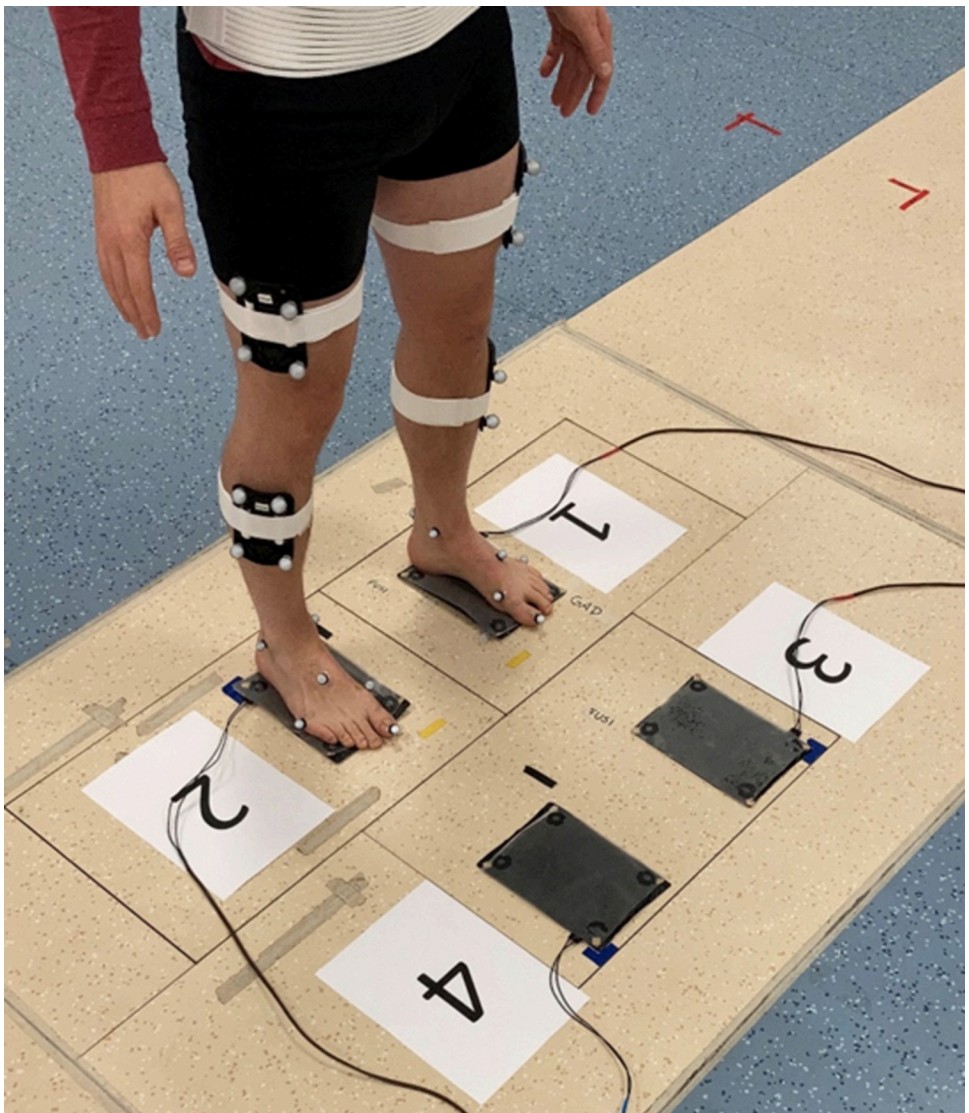

**Fig 1. Experimental setup of the pressure-sensitive target mats used in Experiments 1 and 2.** The spatial layout and configuration was identical in both experiments. Reflective markers for the optoelectronic motion capture system were placed on participants' feet and lower extremities in Experiment 2 only.

step onto these target fields with both feet according to the sequence as given by the auditory stimuli. Fig 1 shows the square configuration of four pressure-sensitive mats. The pressure-sensitive mats provided response latencies using a customized response-acquisition system (1000 Hz). The mats on participants' left side were marked with an odd number (1, 3), while the mats on the right were marked with an even number (2,4). This arrangement was compatible with the "markedness of response codes" (MARC) effect [46]. Participants' starting position at the beginning of a block was with the left foot on mat 1 and with the right foot in mat 2.

## Design and outcome parameters

Predictability of the stimulus sequence (random vs. predictable sequence) was the independent within-subject variable. By means of the pressure-sensitive mats, participants' percent error

and the duration when the pressure-sensitive target mat (T-TARGET) was triggered by a step were extracted from any stepping responses detected. In order to determine any sequence-specific learning effects within participants, the extent of the increase in response times and error rates from Block 8 to Block 9 and the decrease from Block 9 to 10 was interpreted as a measure of how well the respective sequence was internalized. Thus, the sequence-specific learning effect was determined as the difference between the average of Blocks 8 and Blocks 10 and performance in the random Block 9.

## Statistical analysis

The first trial of each block was excluded from analysis. In addition, for the calculation of the average response latencies, both any error trials and correct trials following an error trial were excluded to avoid the influence of post-error slowing on latencies. All statistical computations were performed in R Studio 1.1.456. Paired t-tests were applied to test for a significant change in the observed learning effects against a hypothetical learning effect of 0. An alpha level of 5% was used to determine statistical significance. In order to determine a group average learning rate, we fitted an exponential decreasing non-linear regression function $x(t) = C + A^* e^{\left(-\frac{t}{B}\right)}$ to the averaged performance curves across those blocks with a non-random sequence (Blocks 3 to 8 and Block 10) from which we obtained the function parameters A (intercept), B (time constant) and C (asymptote). For a considerable number of individual participants fitting the exponential function did not provide adequate fits (explained variance < 0.75%) so that the function fitting was conducted on a group level only.

In order to investigate if the sensorimotor complexity of a specific step interacted with learning effect, each step was classified in terms of its direction (forward step, backward step, step with the same foot on the spot) and whether a step was performed with the same or different leg with respect to the stepping leg in the previous trial. Both target sequences did not allow a step on the spot, when the previous step was performed with the same foot. Thus, instead of six different step types, only five step types could occur in each sequence. In an ad hoc ordering, (1) stepping backwards with a different foot than previously was considered the most complex as it required to shift body weight to the other foot and take a step without visual feedback of the stepping target. On a continuum between these two ends from more to less complex, we ordered (2) stepping on the spot following a different foot, (3) stepping forward following a different foot, and (4) stepping backward with the same foot as the previous step. Finally, we considered (5) stepping forward with the same leg as in the previous trial the least complex. For the learning effects of both primary outcome parameters, stepping complexity was included in separate ANOVAs as a 5-level within-subject factor and age group as between-subject factor. Bonferroni-adjusted single comparisons were performed between step complexity conditions, where necessary.

## Results and discussion

Of the 22 participants 17 (77%) reported the subjective impression that some regularities in the sequence were present but only three (14%) individuals were able to report a correct partial sequence of at least 4 elements with a maximum of 9 elements. Table 1 summarizes the descriptive statistics and statistical results for the paired t-tests and the parameters for the exponential learning curves across the 7 blocks containing repeated sequences for Experiment 1.

The response latency in terms of the time point after stimulus onset, at which a step triggered the pressure-sensitive target mat (T-TARGET), showed a strong sequence-specific learning effect (t(21) = 4.58, p < 0.001, dz = 0.98). A strong learning effect was also found for the

**Table 1. Descriptive statistics, sequence-specific learning effects and performance curve parameters for the participants in Experiment 1.**

| | Mean Blocks 8 and 10 (SD) | Mean Block 9 (SD) | Learning effect (AV, SD) | Learning effect statistical comparisons | Intercept | Asymptote | Time constant |
|---|---|---|---|---|---|---|---|
| | Young (n = 22) | Young (n = 22) | Young (n = 22) | All vs. 0 (df = 21) | Young (n = 22) | Young (n = 22) | Young (n = 22) |
| PE (%) | 1.83 (1.33) | 4.23 (2.62) | 2.40 (2.29) | **t = 4.92, p < 0.001 dz = 1.05** | 3.39 | 1.68 | 0.38 |
| T-TARGET (ms) | 892 (142) | 954 (152) | 56 (57) | **t = 4.58, p < 0.001 dz = 0.98** | 943 | 897 | 0.94 |

PE: Percent error.

percent errors (PE) (t(21) = 4.92, p < 0.001, dz = 1.05). Fig 2A and 2B show the average performance of all participants across the 10 blocks.

Step complexity affected the learning effect in terms for percent error (F(1,4) = 4.78, p = 0.002, partial eta^2 = 0.19). Post-hoc single comparisons indicated that stepping forward with a different foot (lowest learning effect: mean = -1.1%, SD 2.2) showed the opposite tendency to stepping backwards with a different foot (greatest learning effect: mean = 4.4%, SD 5.1; p < 0.001) and stepping backwards with the same foot (mean = 2.7%, SD 4.0; p = 0.004), both conditions of which showed positive learning effects. A similar, numerical tendency was observed for stepping on the spot with a different foot (mean = 2.9%, SD 5.5; p = 0.08). In contrast, T-TARGET was not was affected by step complexity (F(4,84) = 0.38, p = 0.82, partial eta^2 = 0.02). All step complexity conditions showed positive learning effects with the lowest effect when stepping forwards with the same foot (mean = 28.1 ms, SD 110.0) and the greatest learning effect when stepping backwards with the same foot (mean = 51.7 ms, SD 99.0).

The above results demonstrate that sequence learning is expressed in the context of a more complex task, such as goal-directed stepping. The strong effect sizes for both the response latency and the error rate indicate that the sensorimotor complexity of the motor task did not diminish the acquisition of a sequence. The statistical structure of the sequences used in our study, in terms of their sequential transitions, were the same as in Zirngibl and Koch [10]. Therefore, it appears as if the rate of individuals with a high degree of explicit knowledge of the sequences was considerably lower in this experiment compared to the data reported in Zirngibl and Koch [10], where 28% to 35% of the participants acquired a high degree of explicit knowledge of at least 5 correct elements depending on the response mode (verbal > manual).

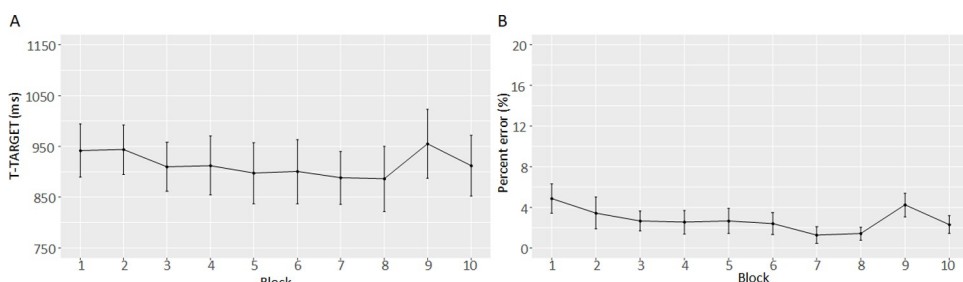

**Fig 2.** Performance curves of the (A) stepping response latencies and (B) percent error. In the Blocks 1, 2 and 9, the sequence of stimuli was random, while in the remaining Blocks 3 to 8 and 10 the respective target sequence was presented 6 times. T-TARGET: time point step onto pressure-sensitive mat. Error bars represent 95% confidence intervals.

It is important to consider, however, that Zirngibl and Koch [10] used manual and vocal responses and therefore imposed different motor demands. It is possible, therefore, that the demands of the locomotor task imposed a processing load, which prevented the knowledge of the sequence to reach a conscious level.

In Experiment 2, we used our novel locomotor sequence task to investigate if sensorimotor and cognitive changes associated with normal ageing would interfere with sequence learning. In addition, we were interested in subdividing the stepping responses into its subcomponents in order to assess the earliest stage of movement execution at which a sequence-specific learning effect could be found. This would provide us further insights into the movement parameters that would be influenced selectively by the acquired sequence knowledge.

## Experiment 2

The major aim of this experiment was to investigate age-related differences in locomotor sequence learning. The basic methodology of this experiment was the same as described in Experiment 1. In the following, we will describe how Experiment 2 extended the methodological approach of Experiment 1. Older adults were expected to demonstrate limitations in their stepping performance, which would restrict their acquisition of the specific sequences.

## Methods

### Participants

The age inclusion criterion for the younger adults was an age between 18 and 35 years and an age between 65 and 80 for the older adults. The experiment included 27 younger adults (13 females and 14 males; average age = 25.9 +/- 3.9 years, weight = 69.2 +/- 12; height = 173.1 +/- 10.1; Body Mass Index = 23.1 +/- 3.2) and 22 older adults (13 females and 9 males; average age = 72.8 +/- 4.8 years, weight = 74.3 +/- 11.6; height = 166.2 +/- 9.4; Body Mass Index = 26.8 +/- 2.9) who were recruited from the general public. Like Experiment 1, Experiment 2 was reviewed and approved by the ethics committee of the RWTH Aachen University (EK 322/19). All participants provided written informed consent before inclusion in the study according to the Declaration of Helsinki.

Sample sizes in more recent publications that studied aging effects on sequence learning in the upper limb (fingers, hand, arm), especially studies reporting reduced [47–49] or even enhanced [31] learning performance in older adults above 65 years, ranged from 15 up to 40 participants per age group. Therefore, we expected that a sample size of at least 22 participants per age group would provide sufficient statistical power to discover large age effects. Based on the recruited sample sizes, we performed a post-hoc power calculation [G*Power 3.1.9.7; 50] to determine the hypothetical power for finding a difference between the two groups in terms of their learning effects. This resulted in a statistical power of 0.77 for finding a large between-group effect size (dz = 0.8) at an alpha of .05.

All older adults completed a survey regarding information on health and lifestyle and were assessed with the Falls Efficacy Scale, the history of the fall and the german adaptation of the Activities-specific Balance Confidence scale (Mini Mental State Examination = 28.6 +/- 1.3; Falls Efficacy Scale = 18.2 +/- 2.5; Activities-specific Balance Confidence = 96.2 +/- 3.9). No participant reported any symptoms of acute pain during the testing session. Generally, older participants were recruited from a research database of the geriatric hospital hosting the gait lab. No participants were included, if a diagnosis of a neurological diseases such as apoplexy, Parkinson's disease, multiple sclerosis, epilepsy, or a rheumatological or autoimmune disease with an acute attack and/or therapy with antibody therapy were reported. All participants had to show free movement of the upper and lower extremities, which had to be possible without

pain (e.g. due to osteoarthritis) and no apparent cognitive deficit [$\leq$ 24 points in Mini-Mental State Examination; 51]. As additional exclusion criteria, participants were not tested if they reported a severe visual impairment (e.g. acute glaucoma attack, blindness on both sides or an unstable, wet macular degeneration), a severe hearing impairment (e.g. severe hearing impairment, residual hearing impairment and deaf), or acute, exacerbated Chronic Obstructive Pulmonary Disease, uncontrolled cardiovascular disorders (e.g. acute cardiac decompensation with New York Heart Association stage 4, no recent heart attack), or dependency on walking aids such as rollators, walking sticks.

## Experimental setup and procedure

In addition to the pressure-sensitive mats used in Experiment 1, a motion capture system was included consisting of a 10 camera-based optoelectronic system (Qualisys Medical AB, Oqus 500+, Gothenburg, Sweden; 120 Hz). The presentation of the auditory stimuli was recorded as stereo signals in 2 additional analog channels of the motion capture system. This enabled segmentation of the motion capture recordings of a single block into the constituting individual stimulus trials.

In order to enable kinematic analysis of participants' steps in terms of their 3D trajectories, passive, hyper-reflective markers were placed at anatomically predefined locations on the lower extremities as well as on the pelvis and upper extremities of all participants. Clusters of 4 markers were mounted on rigid plastic plates, which were placed on each segment of the lower extremities. The kinematic and kinetic data were processed in MATLAB (2019b, The Math-Works, Inc., Natick, USA). Kinematic data were spline interpolated from 120 Hz to 2400 Hz and subsequently merged with the kinetic data. All timeseries data were smoothed using a generic dual-pass 4$^{th}$-order Butterworth lowpass filter (10 Hz cut-off).

## Design and outcome parameters

The predictability of a stimulus sequence (random vs. predictable sequence) served as an independent within-subject variable and age group was included as an independent between-subject variable (younger adults vs. older adults). In addition to participants' percent error, several types of response latencies were determined based on stepping responses detected in the two data modalities of the pressure-sensitive mats and the foot kinematics. (i) The earliest response event that was extracted was the initiation of a step detected by the motion onset of a foot marker (T-STEP; 3 SD velocity threshold above baseline). This was followed by the (ii) time point when the step onto the pressure-sensitive target mat (T-TARGET) was completed. In between these events, additional movement parameters such as the step amplitude (STEP-AMP), step duration (STEP-DUR) and the time point (T-PVEL) and magnitude of peak velocity (M-PVEL) of the step trajectory were extracted to better understand sequence learning-related adjustments in the control of the stepping movements. Fig 3 shows illustrative traces indicating the three major motion events in a single forward step of the left foot of a single participant.

## Statistical analysis

As in Experiment 1, the first trial of each block was excluded from analysis. In addition, for all outcome parameters except percent error, error trials and the trials following an error trial were excluded to avoid the influence of post-error slowing on latencies and an alpha level of 5% was used to determine statistical significance. We used a 1-dimensional Chi^2 test with Yates correction to assess the significance of the frequencies of correctly reported partial sequences in the group of older adults based on expected frequencies derived from the

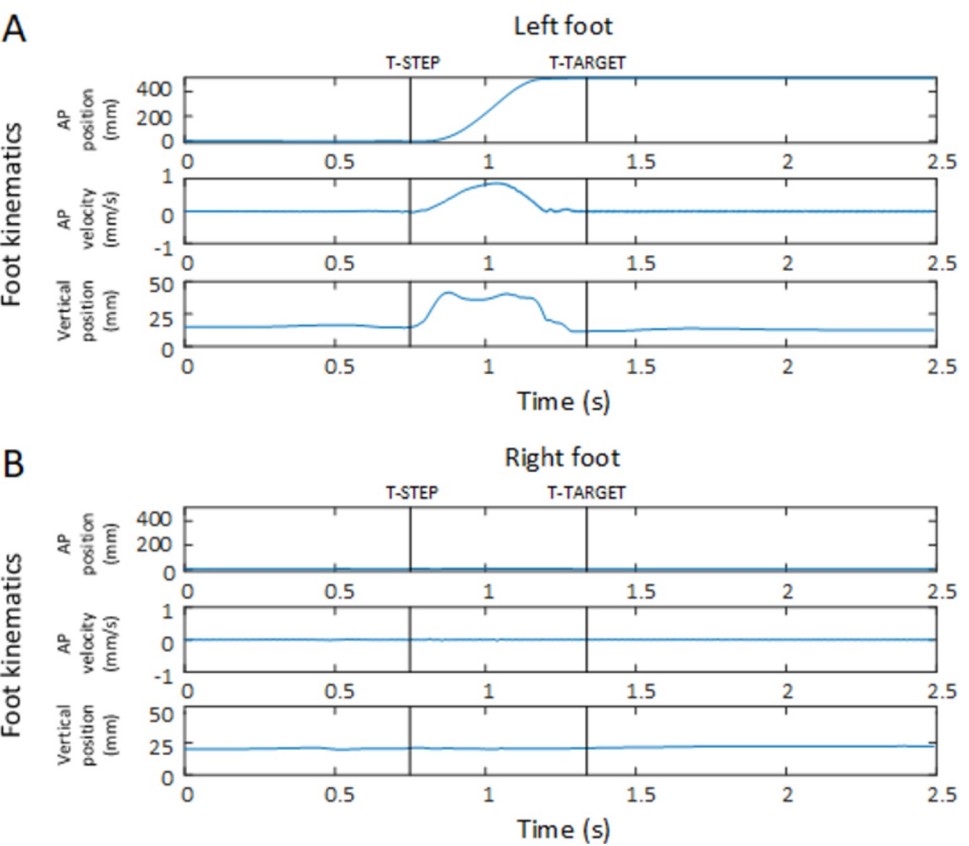

**Fig 3.** Illustrative traces indicating the three major motion events in a single forward step of the (A) left foot of a single participant. During this step, the (B) right foot remains static, while the left foot is stepping forward. T-STEP: time point step onset; T-TARGET: time point step onto pressure-sensitive target mat; AP: anteroposterior.

frequencies observed in the group of younger adults. The critical Chi^2 value for significance was selected for testing the unidirectional alternative hypothesis that older adults would perform worse than the younger adults.

As it was possible that a participant performed more than one step in response to a single stimulus and as the pressure-sensitive mats registered a single step following a stimulus only, stepping sequences detected by the pressure-sensitive mats were manually compared to the stepping sequences detected by the motion capture system. All trials in which the detected steps could not be matched between the two data modalities based on runs of three consecutive steps were excluded from data analysis. Thus, 21% of all trials could not be assigned with certainty and therefore were excluded from the statistical analysis.

As in Experiment 1, paired t-tests were applied to test for a significant change in the observed learning effects within participants against a hypothetical learning effect of 0, while differences between the two age groups were also tested by calculation of two-sample t-tests. In addition, mixed repeated-measure analyses of variance (ANOVAs) with sequence condition as within-subject factor (average of sequence Blocks 8 and 10 vs. random Block 9) and age group as between-subject factor were calculated for the primary outcome parameters T-STEP, T-TARGET.

Linear regressions and Pearson correlations were calculated between all performance parameters and participants' person specific characteristics in order to detect interdependencies and the specificity of learning effects. In order to protect against the problem of multiple

testings, a conservative alpha level of 0.1% was applied for statistical significance. As in Experiment 1, for a considerable number of individual participants fitting the exponential function did not provide adequate fits (explained variance < 0.75%) so that the function fit was conducted on a group level.

In order to investigate if the sensorimotor complexity of a specific step interacted with age group or learning effect, each step was classified as described for Experiment 1. For the learning effects of the all outcome parameters, stepping complexity was included in separate mixed ANOVAs as a 5-level within-subject factor and age group as between-subject factor. Bonferroni-adjusted single comparisons were performed between step complexity conditions where necessary.

## Results and discussion

After being asked if they detected any regularities in the sequence of target stimuli or could recall parts or the entire sequence, 19 (71%) of the younger and 15 (68%) of the older participants declared that they felt that regularities in the sequence were present. The frequency of this subjective impression did not differ between age groups ($Chi^2(1) = 0.06$, $p > 0.1$). Of the younger adults, 6 (22%) reported a partial sequence of at least 4 elements correctly, while 2 individuals (7%) were able to report the full sequence. Of the older adults, 2 individuals (9%) reported a correct partial sequence between 4 and 6 items only. Nevertheless, the relative proportion of conscious recall of the sequence was not different between both age groups ($Chi^2(1) = 1.62$, $p > 0.1$). Also in terms of the average number of elements recalled, a difference between the young adults (mean = 1.68, SD 3.46) and the older adults (mean = 0.64, SD 1.71; t (41.19) = 1.39, p = 0.17) was not observed.

We observed general slowing and generally less accurate performance in the group of older participants but no group differences in sequence-specific learning. Fig 4 shows the performance curves across the 10 practice blocks of the experiment for the three major response stages and percent error as function of age group. Table 2 provides the function parameters for the performance curves across the 7 blocks with repeated sequences (Blocks 3 to 8 and 10). The function intercepts and asymptotes of the performance curves indicate general slowing in the older adults but function slopes were similar between the two age groups for the latency of stepping onto a target mat and the latency of step initiation. Overall, the function slopes indicate that just a single block of exposure resulted in at least 37% of the final performance improvements.

In the final three blocks (Blocks 8, 9, and 10), the latencies of a step onto the pressure-sensitive target mat (T-TARGET) showed a main effect of age group ($F(1,47) = 8.27$, $p = 0.006$, partial $eta^2 = 0.15$), with the young adults responding faster than the older adults, and a main effect of sequence condition ($F(1,47) = 35.35$, $p < 0.001$, partial $eta^2 = 0.43$), with faster responses in the sequence blocks than in the random block. Clearly, no interaction between the two factors was present ($F(1,47) = 0.0008$, $p = 0.98$, partial $eta^2 < 0.001$). The same pattern occurred for the onset latency of a step (T-STEP; group effect: $F(1,47) = 13.74$, $p < 0.001$, partial $eta^2 = 0.23$; sequence condition: $F(1,47) = 21.42$, $p < 0.001$, partial $eta^2 = 0.31$; interaction: $F(1,47) = 1.75$, $p = 0.19$, partial $eta^2 = 0.04$) and the percent error (group effect: $F(1,47) = 13.41$, $p < 0.001$, partial $eta^2 = 0.22$; sequence condition: $F(1,47) = 4.56$, $p = 0.04$, partial $eta^2 = 0.09$; interaction: $F(1,47) = 0.71$, $p = 0.41$, partial $eta^2 = 0.01$) with less errors in younger adults and when the stimuli were predictable. Fig 5 shows the response latencies for the three major response stages and percent error for the final three blocks as a function of age group and sequence condition.

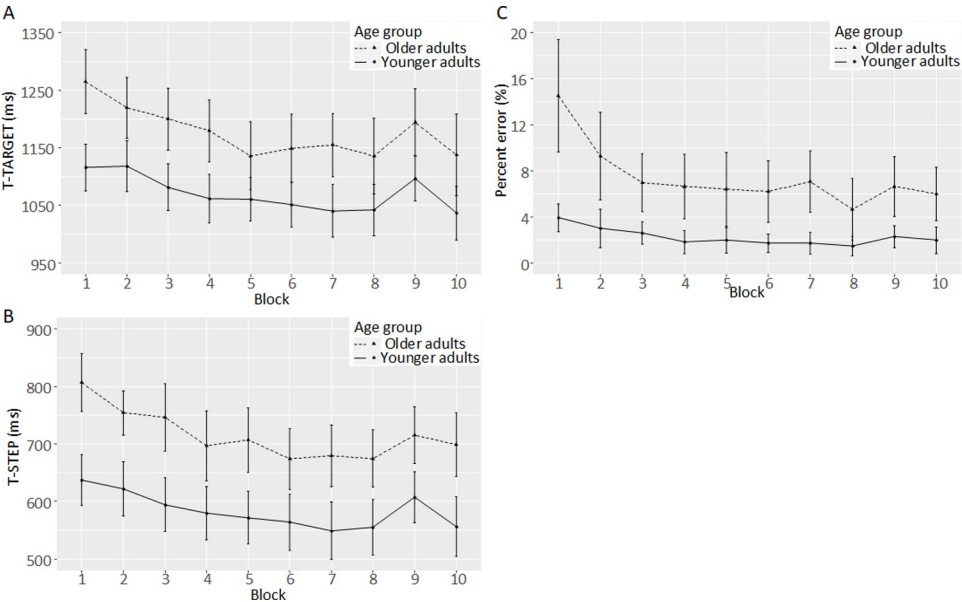

**Fig 4.** Performance curves of the (A and B) response latencies for the main stepping events and (C) percent error. In the Blocks 1, 2 and 9, the sequence of stimuli was random, while in the remaining Blocks 3 to 8 and 10 the respective target sequence was presented 6 times. T-STEP: time point step onset; T-TARGET: time point step onto pressure-sensitive mat. Error bars represent 95% confidence intervals.

Performance reductions in the random Block 9 compared to the sequence Blocks 8 and 10 are clearly visible for the latency of stepping onto the target mat, the latency of step initiation, and the percent error (Figs 4 and 5). Table 3 summarizes the descriptive statistics and statistical results of the t-tests for all movement parameters extracted. All movement parameters except for the stepping duration showed significant sequence-specific learning effects (all $t(48) > 2.07$, all $ps < 0.04$, all $dz > 0.30$). Acquisition of the sequences resulted in reduced latencies of step initiation and stepping on the target mat, longer stepping amplitudes, higher peak velocities during the swing phase at an earlier time point, and an earlier peak velocity of the lateral CoP shift. No differences in sequence-specific learning effects between the age groups were found (all unsigned $t$(all $df>40$) $< 1.54$, all $ps > 0.13$, all $d < 0.45$; Table 3).

Across the entire group of participants (n = 49), the sequence-specific learning effects in all temporal parameters showed strong correlations, while learning effects in movement parameters such as step duration, step amplitude and peak velocity during stepping were less strongly or uncorrelated (Table 4). Additional linear regression analyses between the sequence-specific learning effects in the two main movement parameters (T-STEP, T-TARGET) and individual characteristics of a person such as age, Body Mass Index, and the Mini Mental State Examination, Falls Efficacy Scale and Activities-specific Balance Confidence scores in the group of older adults did not show any correlations between learning effects and parameters (Table 5).

**Table 2. Performance curve parameters for the participants in Experiment 2.**

|  | Intercept | | Asymptote | | Time constant | |
|---|---|---|---|---|---|---|
|  | Young (n = 27) | Old (n = 22) | Young (n = 27) | Old (n = 22) | Young (n = 27) | Old (n = 22) |
| PE (%) | 5.38 | 6.34 | 3.99 | 3.12 | 0.67 | 0.94 |
| T-TARGET (ms) | 1116 | 1224 | 1038 | 1133 | 0.53 | 0.43 |
| T-STEP (ms) | 622 | 761 | 551 | 679 | 0.46 | 0.52 |

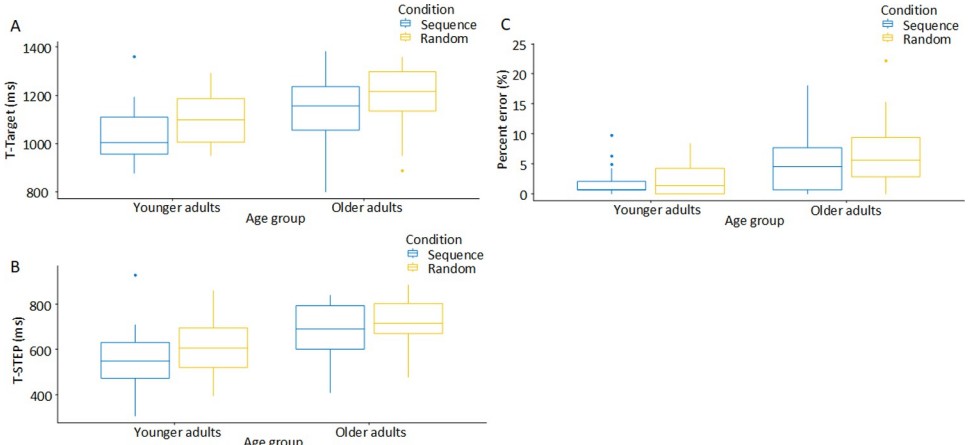

**Fig 5.** Box plots for the (A and B) latencies for the main stepping events and (C) percent error as a function of age group and the sequence condition for the final three blocks (average of sequence Blocks 8 and 10 vs random Block 9). T-TARGET: time point step onto pressure-sensitive mat; T-STEP: time point step onset. The whiskers of the boxes indicate variability outside the upper and lower quartiles. Relative outliers may be plotted as individual points.

The statistical results for the outcome parameters regarding an effect of stepping complexity on the learning effects are presented in Table 6. In general, ANOVAs for all specific outcome parameters on the learning effects in terms of the difference between the random block 9 and the average of sequence blocks 8 and 10 did not find any interactions between age group and step complexity. T-STEP showed the tendency for an effect of complexity (F(4,188) = 2.24, p = 0.09, partial eta^2 = 0.05), while T-TARGET was affected by complexity (F(4,188) = 3.52, p = 0.01, partial eta^2 = 0.07). Neither of the two parameters showed an effect of age group

**Table 3. Descriptive statistics, age comparison and sequence-specific learning effects in Experiment 2.**

| | Mean Blocks 8 and 10 (SD) | | Mean Block 9 (SD) | | Sequence-specific learning effect (AV, SD) | | Sequence-specific learning effect statistical comparisons | | | |
|---|---|---|---|---|---|---|---|---|---|---|
| | Young (n = 27) | Old (n = 22) | Young (n = 27) | Old (n = 22) | Young (n = 27) | Old (n = 22) | Young vs. old | All vs. 0 (df = 48) | Young vs. 0 (df = 26) | Old vs. 0 (df = 21) |
| PE (%) | 1.70 (2.28) | 5.33 (5.26) | 1.89 (2.21) | 6.26 (6.05) | 0.56 (2.10) | 1.30 (3.89) | t = -0.79 p = 0.43 d = 0.24 | **t = 2.07, p = 0.04 dz = 0.30** | t = 1.39 p = 0.18 dz = 0.27 | t = 1.56 p = 0.13 dz = 0.33 |
| T-TARGET (ms) | 1039 (113) | 1136 (150) | 1096 (98) | 1195 (131) | 58 (68) | 58 (68) | t = -0.03 p = 0.98 d = 0.008 | **t = 6.05, p < 0.001 dz = 0.86** | **t = 4.42 p < 0.001 dz = 0.85** | **t = 4.04 p < 0.001 dz = 0.86** |
| T-STEP (ms) | 556 (124) | 687 (116) | 607 (112) | 717 (112) | 52 (59) | 29 (63) | t = 1.31 p = 0.20 d = 0.38 | **t = 4.75, p < 0.001 dz = 0.68** | **t = 4.59 p < 0.001 dz = 0.88** | **t = 2.14 p = 0.04 dz = 0.46** |
| STEP-AMP (mm) | 404 (19) | 379 (37) | 336 (30) | 323 (40) | -69 (30) | -58 (32) | t = -1.21 p = 0.23 d = 0.35 | **t = -14.54, p < 0.001 dz = 2.08** | **t = -12.11 p < 0.001 dz = 2.33** | **t = -8.52 p < 0.001 dz = 1.82** |
| STEP-DUR (ms) | 706 (129) | 744 (128) | 701 (143) | 744 (126) | -6 (56) | 2 (47) | t = -0.50 p = 0.62 d = 0.14 | t = -0.31, p = 0.76 dz = 0.04 | t = -0.52 p = 0.61 dz = 0.10 | t = 0.18 p = 0.86 dz = 0.04 |
| T-PVEL (ms) | 863 (149) | 1023 (149) | 921 (144) | 1055 (124) | 58 (68) | 31 (77) | t = 1.32 p = 0.20 d = 0.38 | **t = 4.43, p < 0.001 dz = 0.63** | **t = 4.45 p < 0.001 dz = 0.86** | t = 1.88 p = 0.07 dz = 0.40 |
| M-PVEL (mm/s) | 856 (201) | 794 (136) | 746 (134) | 735 (142) | -113 (105) | -60 (129) | t = -1.54 p = 0.13 d = 0.45 | **t = -5.28, p < 0.001 dz = 0.75** | **t = -5.59 p < 0.001 dz = 1.08** | **t = -2.18 p = 0.04 dz = 0.46** |

**Table 4. Correlations between sequence-specific learning effects of all movement parameters for the group of all participants in Experiment 2.**

|          | T-TARGET | T-STEP | STEP-DUR | STEP-AMP | T-PVEL | M-PVEL |
|----------|----------|--------|----------|----------|--------|--------|
|          | R, p     |        |          |          |        |        |
| T-STEP   | **0.66, <0.001** |  |      |          |        |        |
| STEP-DUR | 0.37, 0.008 | 0.05, 0.74 |   |          |        |        |
| STEP-AMP | -0.15, 0.30 | -0.21, 0.14 | -0.02, 0.87 |     |        |        |
| T-PVEL   | **0.71, <0.001** | **0.88, <0.001** | 0.30, 0.03 | -0.17, 0.24 |  |        |
| M-PVEL   | 0.23, 0.12 | 0.16, 0.28 | 0.18, 0.23 | 0.11, 0.47 | 0.27, 0.06 |  |

(age group: both $F(1,47) < = 2.52$, both $ps > = 0.12$, both partial eta^2 = 0.05) or an interaction between age group and step complexity (both $F(4,188) < = 1.21$, both $ps > = 0.31$, both partial eta^2 = 0.03). Post-hoc single comparisons for T-TARGET indicated that the learning effect tended to differ only between stepping on the spot following a step of the other foot (lowest learning effect: mean = 33.9 ms, SD 126.1) and taking a backward step following a step with the same foot (greatest learning effect: mean = 100.7 ms, SD 131.6; p = 0.08). The learning effects in STEP-AMP, T-PVEL and M-PVEL were influenced by step complexity as well. For STEP-AMP, single comparisons indicated the stepping forward with the same foot resulted in the strongest learning effect compared to all other step complexities (mean = -53 mm, SD 105; all p < 0.01; Fig 6A). For the other stepping complexities, learning effects were not different from 0. Another pattern was observed for M-PVEL (Fig 6B). A learning effect when stepping forward with the same foot (mean = 188 mm/s, SD 418) as well as stepping backward with a different foot (mean = 155 mm/s, SD 259) meant an increase in the peak velocity. No learning effect was observed for stepping in the spot with a different foot (mean = -0.29 mm/s, SD 319), while stepping forward with a different foot (mean = -99.8 mm/s, SD 188) and stepping backward with the same foot (mean = -396 mm/s, SD 511) lead to reductions in peak velocity. Fig 6 summarizes the effect of stepping complexity on learning effect.

The latencies for T-TARGET appeared slower for the young adults in Experiment 2 compared to the young adult participants in Experiment 1, even though the sequence and the task requirements were completely similar. Therefore, a mixed analysis of variance with experiment as between-subject factor and the series of all blocks as within-factor was computed, which confirmed a significant effect of block ($F(9,423) = 16.03$, p<0.001, partial eta^2 = 0.25) with respect to the generally observed decreasing performance curves and faster responses for the participants who took part in Experiment 1 ($F(1,47) = 24.63$, p<0.001, partial eta^2 = 0.34). An interaction between both factors was not found ($F(9,423) = 1.39$, p = 0.23, partial eta^2 = 0.03). The percent error did not differ between the groups of young adults in Experiments 1 and 2. Only an effect of block was found ($F(9,423) = 8.30$, p<0.001, partial eta^2 = 0.15; group: $F(1,47) = 1.23$, p = 0.27, partial eta^2 = 0.03; interaction: $F(9,423) = 1.09$, p = 0.37, partial eta^2 = 0.02). As a motion capture system was not available for Experiment 1, no other motion parameters could be compared between both groups. Hence, the young

**Table 5. Correlations between sequence-specific learning effects and individual characteristics of the older adults in Experiment 2.**

|               | Age | | BMI | | MMSE | | FES | | ABC-D | |
|---------------|-----|---|-----|---|------|---|-----|---|-------|---|
|               | R² (%) | p | R² | p | R² | p | R² | p | R² | p |
| T-STEP (ms)   | <0.001 | 0.99 | 0.02 | 0.95 | 10 | 0.14 | 4 | 0.35 | 1.8 | 0.56 |
| T-TARGET (ms) | 16 | 0.06 | 3 | 0.41 | 5 | 0.34 | 5 | 0.34 | 12 | 0.11 |

BMI: Body Mass Index; MMSE: Mini Mental State Examination; FES: Falls Efficacy Scale; ABC-D: German adaptation of Activities-specific Balance Confidence scale.

**Table 6. Effect table for the effect of step complexity and age group on sequence learning effects.**

| | PE (%) | T-TARGET (ms) | T-STEP (ms) | STEP-AMP (mm) | STEP-DUR (ms) | T-PVEL (ms) | M-PVEL (mm/s) |
|---|---|---|---|---|---|---|---|
| Step complexity (dfn = 4, dfd = 188) | F = 1.25, p = 0.29, partial eta^2 = 0.03 | **F = 3.52, p = 0.01, partial eta^2 = 0.07** | **F = 2.24, p = 0.09,** partial eta^2 = 0.05 | **F = 12.23, p < 0.001, partial eta^2 = 0.21** | **F = 2.50, p = 0.06,** partial eta^2 = 0.05 | **F = 3.29, p = 0.02, partial eta^2 = 0.07** | **F = 20.89, p < 0.001, partial eta^2 = 0.31** |
| Step complexity x age group (dfn = 4, dfd = 188) | F = 2.08, p = 0.11, partial eta^2 = 0.04 | F = 1.21, p = 0.31, partial eta^2 = 0.03 | F = 0.77, p = 0.52, partial eta^2 = 0.02 | F = 0.18, p = 0.69, partial eta^2 = 0.004 | F = 1.35, p = 0.26, partial eta^2 = 0.03 | F = 0.35, p = 0.81, partial eta^2 = 0.007 | F = 1.62, p = 0.2, partial eta^2 = 0.03 |

participants in Experiment 1 were generally faster than those in Experiment 2, but the important result is that the learning effects were very similar across the young adults in Experiment 1 and 2. The settings differed between the first and second experiment, but it does not seem as if

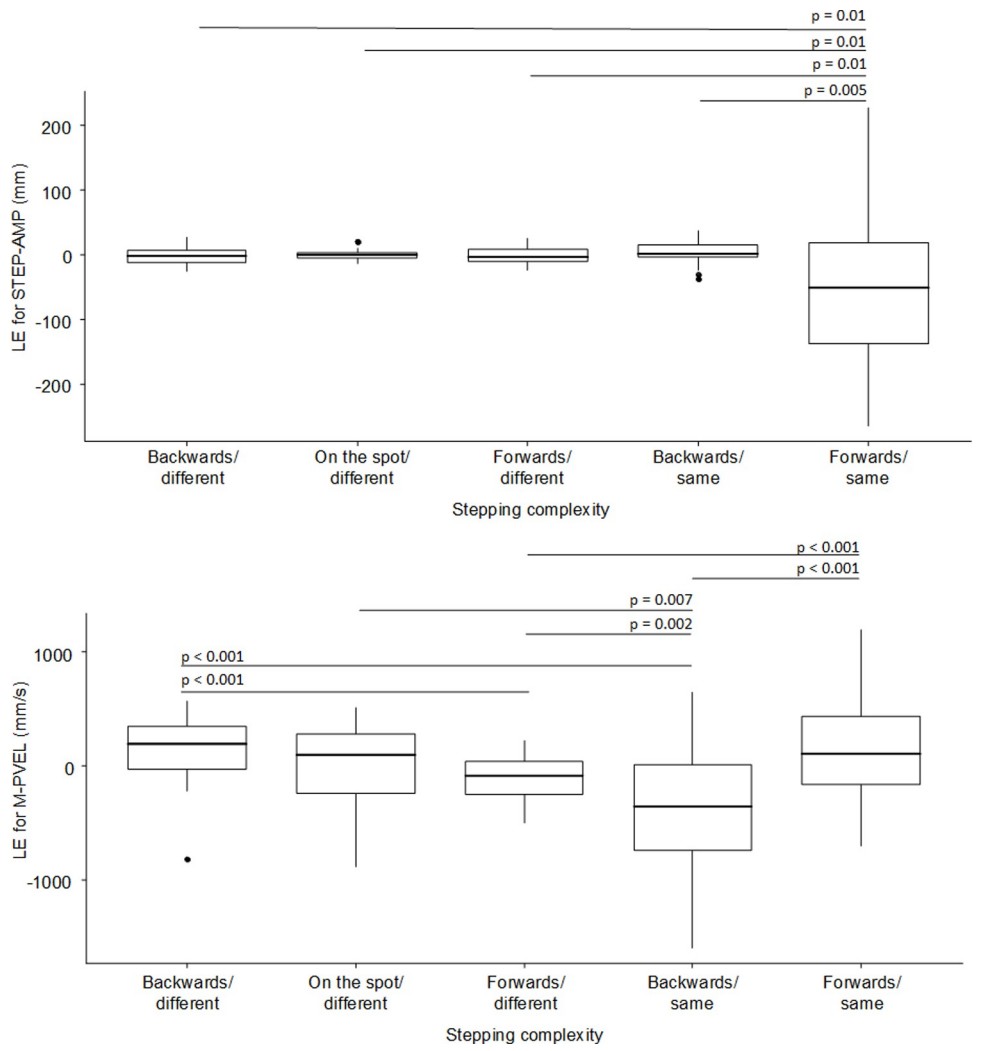

**Fig 6.** Box plots of the sequence learning effects for stepping amplitude (A; STEP-AMP) and average peak stepping velocity (B; M-PVEL) as a function of the direction of a step and the foot of the previous step. The whiskers of the boxes indicate variability outside the upper and lower quartiles. Relative outliers may be plotted as individual points.

the changed environmental context of the gait lab setting emphasized accuracy of stepping over speed in the younger adults in Experiment 2 as the two groups did not differ in terms of the progression of percent error. The young adults in Experiment 2 were slightly older and had been recruited from the general public, while the younger adults in Experiment 1 were recruited from a body of psychology students and therefore possibly more experienced in reaction-time paradigms. The more extensive experimental procedure in Experiment 2 might also have motivated the participants to adopt a more conservative trade-off between the response latencies and the spatial stepping accuracy according to Fitts law [longer latencies with less variable step positions; 52]. A direct comparison between Experiment 1 and 2 regarding the spatial variability of the stepping was not possible due to the aforementioned lack of a motion capture system in Experiment 1.

The results of Experiment 2 demonstrate that the sensorimotor complexity of stepping does not impair procedural learning of sequences in older adults compared to younger adults despite a general slowing in the response latencies and reduced accuracy in the older adults. In terms of the duration of stepping onto the pressure-sensitive target mat (T-TARGET), a backward step following a step with the same foot showed numerically the greatest learning effect in both Experiment 1 and 2. Although the interaction between age group and step complexity was not significant, the average learning effect appeared numerically the greatest in older adults (mean = 133.0 ms, SD 164.0) when stepping backwards with the same foot. That aging does not reduce the capacity for procedural learning of the sequences is also expressed by any of the recorded person-related characteristics, such as age, Body Mass Index, cognitive level, or balance scores, not predicting the sequence-specific learning effects in the older adults. The ability to anticipate the stimuli in a sequence and prepare responses did not only affect the preparation of the steps in terms of the response latencies, but also the characteristics of the executed steps themselves, such as that sequence learning led to earlier, faster and longer steps.

Finally, the fact that the correlations indicated that peak stepping velocity and stepping amplitude were independent factors, however, indicates that sequence knowledge is utilized on different levels of locomotor control of taking a step. This conclusion is also supported by different patterns of the influence of stepping complexity on the two parameters.

## General discussion

We developed a step initiation and goal-directed stepping (locomotor) SRTT paradigm in order to test sequence learning under more complex demands in postural and motor control, such as proactive postural stabilization before step initiation. Our novel paradigm allowed us to delineate the effect of an advanced age (i.e. $\geq$ 65 yrs) on locomotor sequence learning as we expected that age-related performance decrements in balance control and locomotion would impair the acquisition of the regularities inherent in the task. In two experiments, we demonstrated successful motor sequence learning in our goal-directed stepping SRTT. We found clear learning effects with respect to the latencies of two stepping events: the moment of step initiation and the conclusion of the step by the triggering of the target mat. Further parameters of the stepping movement were also adapted in the course of sequence acquisition resulting in faster and longer steps with the predictable sequence, although stepping duration was not altered by learning.

Importantly, despite alterations in sensorimotor function that occur with aging (i.e. chronological age), our results demonstrated that aging did not seem to have an obvious influence on the acquisition of the stepping sequence in terms of any latencies, movement parameters or error rates. Although tending to be more variable in their performance, the older adults demonstrated a comparable amount of learning of the sequences nevertheless, which also was not associated with any clinical measures of balance control and confidence or general cognitive

aptitude. This contrasts with previous observations that implicit sequence learning becomes more limited in older age, especially with more complex sequences [38]. Thus, it may be the case that locomotor sequence learning is less susceptible to the effects of aging than the learning of sequences in other action and effector contexts (e.g. manual key press sequences). Alternatively, it could also be possible that effects of aging on sequence learning are reflecting the possible expertise in finger sequence movements of younger adults compared to less experienced older adults. Of course, it needs to be considered as well that smaller differences in locomotor sequence learning may actually exist between younger and older adults but that our study had not sufficient statistical power and sensitivity to detect these differences.

We did find indications of general slowing in the older adults but this did not affect the learning itself. Although the older adults appeared to be less likely to acquire a conscious, explicit representation of the sequence of stimuli or responses, a statistically significant difference was not found. This observation contrasts with age-related differences in explicit sequence monitoring reported by Caljouw et al. [53] in a postural visuomotor sequence learning task involving body weight shifts. Therefore, it may be that the influence of age on the acquisition of explicit sequence knowledge may be specific to the type of movement task or that our experiment was not sufficiently sensitive to uncover an age effect in the acquired explicit knowledge. Likewise, more subtle differences between the two age groups in their learning could exist but were not detected due to insufficient statistical power to find small or medium effects.

Our findings indicate that the biomechanical and behavioural complexity of a locomotor task such as goal-directed stepping does not impede the acquisition of any regularities in a sequence of stimuli. Du and Clark [18] suggested that the aggregation of regularities in their foot-tapping SRTT resulted from the biomechanical constraints imposed by the task, for example the reoccurrence of target positions harder to reach with a foot tap. The relevance of biomechanical constraints in sequence learning tasks was also indicated for responses in manual sequence learning. Differences in motor chunking between young and older adults in a manual sequence production task was attributed to biomechanical factors, such as increased stiffness of finger joints in older adults, which could be misinterpreted as chunking due to systematically and repeatedly occurring delayed latencies of finger with stiffer joints [47]. In other words, biomechanical constraints in the context of a particular SRTT imposed on participants may hide the expression of sequence learning. Compared to the foot tapping paradigm in Du and Clark's study [18], we consider our SRTT less demanding in biomechanical and balance terms as our participants had to perform individual forward or backward steps only without returning to a starting posture so that phases of potential postural instability while standing on a single leg were shorter.

The observations in our study suggest at least two separable paths how the gradually improving prediction of the sequence shaped participants' stepping behaviour. The first path concerns the ability to anticipate a stimulus and corresponding response and thereby to prepare an adequate step in advance. Correlations between the learning effects in our performance measures indicated that response time differences between the predictable and unpredictable sequences in the final three blocks of trials were strongly associated, starting with the earliest response measures, the timepoint of the anticipatory postural weight shifts. The second path relates to the execution of the stepping movements in terms of their spatio-temporal characteristics. Thus, it seems reasonable to assume that our participants prepared not only for the initiation of a step but also prepared the actual execution of a step, so that the stepping duration remained constant but the velocity and distance increased.

Flanagan and colleagues demonstrated that the sensorimotor control system learns to predict the consequences of an action faster than it optimizes the motor control during the action

[54–56]. Mayr [57] argued for a dissociation between the implicit learning of spatial and non-spatial regularities attended to during sequence learning. Koch and Hoffmann [58] concluded that chunking occurred on the basis of relations between events and Koch and Hoffmann [59] reported that the learning of spatial event sequences is the dominant factor in implicit perceptual and motor learning. Correspondingly, Willingham et al. [60] presented findings suggesting that in implicit motor sequence learning the response location seems to play a more important role than the specific sequence of effectors activated. Thus, also in the locomotor stepping sequence learning of our present experiments, the sequence of location of target mats may be an important cue in addition to sensory effects linked to the sequence of final stance postures or the sequence of activated lower extremity muscles.

We envision that our paradigm could be used as a clinical tool for diagnosing impaired functional neuroplasticity and learning in ill-health older adults. Future research should be conducted to assess if geriatric populations with mild or more severe cognitive impairments are still capable of the observed locomotor sequence learning. It should also be determined if the potential inability to acquire the sequences might be used as a diagnostic indicator of the onset of cognitive deficits in older age. Additionally, our task might serve as a physical activity exercise in still mobile geriatric populations by the implementation of motivating performance feedback. Adapted for follow-up therapy aimed at promoting neural plasticity, it may even become a future approach to stimulate experience-dependent neural plasticity to counter age-related functional limitations and neurodegeneration. Research results into neuroplastic alterations in the human brain that occur during implicit and explicit sensorimotor learning are promising [61]. Therefore, it should be investigated if a clinical application of locomotor sequence learning in the context of neurorehabilitation is warranted.

In conclusion, we assessed locomotor sequence learning in a goal-directed stepping task in younger and older adults and observed learning in several parameters of locomotor performance. We did not observe effects of older age to restrict learning of the regularities embedded in the sequences.

## Acknowledgments

We appreciate the support received by Joeline Schulz and Johannes Quandel during data acquisition and data post-processing.

## Author Contributions

**Conceptualization:** Leif Johannsen, Denise Nadine Stephan, Iring Koch.

**Data curation:** Leif Johannsen, Joao Batista, Doreen Schulze, Thea Laurentius.

**Formal analysis:** Leif Johannsen, Doreen Schulze.

**Funding acquisition:** Leif Johannsen, Joao Batista, Iring Koch, Leo Cornelius Bollheimer.

**Investigation:** Leif Johannsen.

**Methodology:** Leif Johannsen, Erik Friedgen, Denise Nadine Stephan, Iring Koch.

**Project administration:** Leif Johannsen, Thea Laurentius, Leo Cornelius Bollheimer.

**Software:** Erik Friedgen.

**Supervision:** Leif Johannsen, Leo Cornelius Bollheimer.

**Visualization:** Leif Johannsen.

**Writing – original draft:** Leif Johannsen.

**Writing – review & editing:** Leif Johannsen, Erik Friedgen, Denise Nadine Stephan, Joao Batista, Doreen Schulze, Thea Laurentius, Iring Koch, Leo Cornelius Bollheimer.

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
