## [Decision Letter · Decision Letter 0]

19 Oct 2021

PONE-D-21-22593

Keeping in step with the young: chronometric and kinematic data show intact procedural locomotor sequence learning in older adults

PLOS ONE

Dear Dr. Johannsen,

Thank you for submitting your manuscript to PLOS ONE. After careful consideration, we feel that it has merit but does not fully meet PLOS ONE’s publication criteria as it currently stands. Therefore, we invite you to submit a revised version of the manuscript that addresses the points raised during the review process.

In addition to addressing all of the points raised by the two reviewers below, I would also like to raise the following: 

a) While I agree with Reviewer 2 that the intro is well-written, it is my opinion that it would be beneficial to considerably reduce the introduction in length. For example, the SRTT paradigm is a common protocol and can be succinctly and clearly introduced in a sentence or two. Similarly, it would be advantageous to only introduce those concepts that are directly relevant to the current paper (e.g., different sequence representations, chunking, etc. were not assessed here and thus do no warrant such an elaborate introduction). A similar comment can be applied to the abstract. Not necessary here to mention the number of blocks or the specific block numbers used in the computation of learning. 

b) It is unclear as to how results from  Experiment 1 were used for the power computation for Experiment 2. Experiment 1 did not assess age differences; and, ultimately, data from the available literature was used for the power comp (Howard & Howard). Regardless, the two groups then exceeded this desired sample size (27 and 22 in the two groups as opposed to 16 each). So there appears to be no data-driven justification for the precise sample in Experiment 2.

c) The following information (applicable to both experiments) is better fit for the data processing / reduction section: “.. the first trial of each block was excluded from analysis. In addition, for all outcome parameters except error proportions, error trials and the trials following an error trial were excluded to avoid the influence of post-error slowing on latencies”

d) If I understood correctly, the measure T-PWS was not valid for mats 3 and 4. Is this correct? If yes, this no longer seems like a useful dependent measure given that the large number of trials that were discarded.

e) For experiment 2, it would be advantageous to indicate whether the sequence specific learning effects (middle columns; Table 3) are significant WITHIN each age group. The last column collapses across the two groups and shows significant learning effects, but this appears to benefit from the large n (collapsed across young and old) and effects that are disproportionately larger in young (although not sig. different from older). In brief, it does not appear that the older adults exhibit significant learning per se on many metrics. This information would be of interest. Note that this does not take away from the main between-group comparisons, but would be worth specifying nonetheless. 

f) I leave this up to the authors but I am not convinced the methods of the two experiments can not be presented within a single methods section followed by results of the two experiments. 

We look forward to receiving your revised manuscript.

Kind regards,

Bradley R. King

Academic Editor

PLOS ONE

“We appreciate the support received by Joeline Schulz and Johannes Quandel during data acquisition and data post-processing. Funded by the Federal Ministry of Education and Research (BMBF) and the Ministry of Culture and Science of the German State of North Rhine-Westphalia (MKW) under the Excellence Strategy of the Federal Government and the Länder. In addition, this study was supported by the Robert Bosch Stiftung in the context of the “Lehrstuhl für Geriatrie an der Medizinischen Fakultät der RWTH Aachen” (32.5.1140.0009.0).”

Additional Editor Comments (if provided):

Reviewers' comments:

Reviewer's Responses to Questions

**Comments to the Author**

1. Is the manuscript technically sound, and do the data support the conclusions?

Reviewer #1: Yes

Reviewer #2: Partly

2. Has the statistical analysis been performed appropriately and rigorously? 

Reviewer #1: Yes

Reviewer #2: No

3. Have the authors made all data underlying the findings in their manuscript fully available?

Reviewer #1: Yes

Reviewer #2: Yes

4. Is the manuscript presented in an intelligible fashion and written in standard English?

Reviewer #1: Yes

Reviewer #2: Yes

5. Review Comments to the Author

Reviewer #1: Keeping in step with the young: chronometric and kinematic data show intact procedural locomotor sequence learning in older adults

This study examined a novel locomotor variant of the serial reaction time task (SRTT) paradigm, in which young and older adults stepped on targets in response to auditory cues, and analyzed sequence learning and age effects across multiple kinematic measures. Overall, sequence learning was observed for young and older adults, and was not affected by age. Specifically, in Experiment 1, young adults showed lower error rates and reduced time to step into the target zone for sequence blocks relative to random blocks. In Experiment 2, both young and older adults showed sequence learning effects for error rate, time to initiate a step, time to step into the target zone, step amplitude, latency and magnitude of peak step velocity, and latency of peak center-of-pressure shift velocity. None of the learning effects in Experiment 2 were affected by age.

The novel task applied in this study provides a useful extension of the SRTT literature, and aging effects thereon, to a more ecologically valid movement condition. The experiment appears to have been conducted with rigor, and the primary findings appear convincing. My concerns and comments are detailed below.

MAJOR

1) The locomotor SRTT has three different step types (forward step, backward step, foot on same spot), two of which (forward, backward) could have followed a step by the same foot, or by the opposite foot. It seems that these five conditions (forward/same foot, forward/different foot, backward/same foot, backward/different foot, same spot) present different postural and balance concerns and difficulty, which could result in different challenges to older adults. The finger movement SRTT literature demonstrates aging effects increase for higher-order sequence information and other forms of increased complexity (e.g., Howard et al., 2004, Psychology and Aging; Bennett et al., 2007, Journals of Gerontology - Series B); it may be then that in a locomotor SRTT, aging effects become apparent for the more difficult/complex movements. Did step type have any impact on sequence learning, or interact with age effects on sequence learning?

MINOR

2) Line 105: The quick mention of generalization feels out of place in the paragraph on chunking; I would recommend separating these concepts, and perhaps expanding on the generalization point.

3) Line 257: What is meant by “Each participant received an individualized set of three randomized sequences”? I presume that this means the sequences used had the same archetypal format across participants (i.e., A-C-D-B-A-C), but that the mapping of [1,2,3,4] to [A,B,C,D] is what was randomized three times for each participant. If this is instead referring to a loop within the random condition, that should be clarified, and potential predictability driven by that loop would need to be addressed in all analyses. Could you please clarify this point?

4) Line 309-310: This sentence is unclear; I believe it is missing a phrase (“In addition, for response latencies but not ???, error trials and…”)

5) Table 1: It would be good to include the random condition means (Block 9) in this Table for direct comparison with the sequence condition means. Also, assumedly EP means error proportion, but that is not specified- and if the units are % it cannot be a proportion.

6) Line 381: It is good that participants with severe hearing impairments were excluded, but what about non-severe aging-related hearing loss? Given that the cues were auditory, even moderate presbycusis could cause additional cue processing time for older adults that would affect responses in a manner unrelated to movement planning. Were audiometric thresholds obtained for older adults, or was hearing assessed in any other objective manner?

7) Figure 3: What do the red and blue traces in the CoP position time series plots differentiate? I am assuming ML and AP, but it should be labelled. Also, perhaps this is just an aspect of the coordinate system used for measurements, but if this Figure illustrates a forward step by the left foot, why are left foot displacement and velocity negative?

8) Line 471: Though the Chi-squared test did not reach significance, the descriptive statistics suggest an aging effect here. Moreover, for older adults the full range of number-of-elements-reported is given, whereas only the lower bound is mentioned for young adults. Did older and young adults differ on number of sequence elements correctly reported?

9) Line 496: After stating the lack of age x sequence condition interaction, it would be helpful to report the lower bound of the observed p-value (e.g., p > 0.2). This comment applies to all such statements of null effects intended to be meaningful in the Results sections.

10) Line 533: What was the reasoning for considering the T-PWS, T-STEP, and T-TARGET measures to be the main movement parameters? If these are the parameters of primary interest, why were multiple other measures included in analyses too?

11) Line 550: The comma between “participants” and “who” disrupts the meaning of the sentence. There are multiple odd commas like this throughout the manuscript- these should be identified and removed.

12) Line 553: Would be good to remind here that the reason T-STEP and T-PWS are not mentioned is that those measures were not possible in Experiment 1. You mention this later (line 580), but it would be useful here.

13) Line 562: I don’t recall seeing the T-PWS sequence effect being reported as marginal in the Results. I see in Table 3 the p-value for that comparison was 0.17, which is a stretch for including as marginal- the authors do hint here though that the effect was more reliable in young adults alone, which is not reported. Given the lack of age x sequence interaction though, such a breakdown would not be a licensed comparison. This claim should either be better defended, or removed.

14) Line 601: The given interpretation that locomotor sequence learning may be less susceptible to aging decline than finger movement sequence learning is certainly reasonable. An alternate interpretation the authors may want to consider is that the aging effects reported in the finger movement literature may instead reflect a relative expertise of young adults for finger sequence movements, rather than an aging-related decline, and that this aging-related difference in expertise is not observed for stepping movements.

Reviewer #2: The authors present a study that examined the effect of aging on locomotor sequence learning. The study views this topic from a different context by assessing locomotion and asks an interesting and important question. The introduction is well-written and describes the current literature and motivation of the for study well. However, the Results and Discussion sections are a bit confusing to understand and the statistical methods and results are not described in adequate detail. The following comments may help address these issues:

1. It is unclear whether accuracy was emphasized to participants. What instructions were participants provided about the speed and accuracy of their movements? This may affect how quickly and accurately they respond to the auditory stimuli.

2. Was a recall or recognition test used to assess whether participants were able to determine that there was a sequence in the stimuli? It seems so, as some results are described, but the procedure and questions asked should be described in the methods.

3. What were the results of the paired t-tests in Experiment 1 and the ANOVA in Experiment 2? The methods describe these analyses, but they are not reported in the results.

4. What is the purpose of the Pearson correlations? What part of the research question were they trying to answer?

5. As older adults may make more anticipatory postural adjustments, how were they accounted for with respect to the postural adjustments all participants made to get ready to take a step? Was there any difference in these augments between the older and young adults?

6. The authors statement that, “Surprisingly, despite decreased functional reserves and reduced locomotor stability that occurs with aging (i.e. chronological age), it had no obvious influence on the acquisition of the stepping sequence in terms of any latencies, movement parameters or error rates” (lines 594-596) is a bit confusion. It seems from Figure 4 that older adults were overall slower across all blocks and made more errors. Perhaps the ANOVA results would clarify this.

7. The conclusion seems a bit overstated and broad. For example, how can this task be used as a clinical tool? Without assessing the task on a clinical population, it is unclear which parameters would be valuable for assessing a clinical population, especially since the authors did not find any significant differences between the older and young adults.

Minor

8. This is a minor stylistic suggestion. There are many abbreviations used in the manuscript that disrupt the flow of the paper. Some abbreviations are defined, but never referred to again (e.g., AR in line 157) that can be taken out.

6. PLOS authors have the option to publish the peer review history of their article (what does this mean?). If published, this will include your full peer review and any attached files.

Reviewer #1: No

Reviewer #2: No

---

## [Author Response · Author response to Decision Letter 0]

28 Jan 2022

Dear Dr. King and Reviewers,

we kindly appreciate your time for editing and reviewing our submitted manuscript. We have considered your feedback and suggestions and have addressed them as best as possible. In the following overview, we have listed your single recommendations and our responses and corresponding changes made to the manuscript.

Editor comments

1. “While I agree with Reviewer 2 that the intro is well-written, it is my opinion that it would be beneficial to considerably reduce the introduction in length. For example, the SRTT paradigm is a common protocol and can be succinctly and clearly introduced in a sentence or two. Similarly, it would be advantageous to only introduce those concepts that are directly relevant to the current paper (e.g., different sequence representations, chunking, etc. were not assessed here and thus do no warrant such an elaborate introduction).”

Response: the introduction was reduced in length by removal of those paragraphs relating to chunking and sequence representations including a short section referring to research conducted in monkeys.

2. “A similar comment can be applied to the abstract. Not necessary here to mention the number of blocks or the specific block numbers used in the computation of learning.” 

Response: the abstract was slightly shortened by the removal of methodological information but also received an edited conclusion.

3. “It is unclear as to how results from Experiment 1 were used for the power computation for Experiment 2. Experiment 1 did not assess age differences; and, ultimately, data from the available literature was used for the power comp (Howard & Howard). Regardless, the two groups then exceeded this desired sample size (27 and 22 in the two groups as opposed to 16 each). So there appears to be no data-driven justification for the precise sample in Experiment 2.”

Response: the comment in the methods section of experiment 2 regarding power calculations for sample size estimation based on aging effects reported in the literature has be revised and reformulated. We still have to refer to the available literature but restrict our expectations to large effects only. This has also been considered in the general discussion section.

4. “The following information (applicable to both experiments) is better fit for the data processing / reduction section: “.. the first trial of each block was excluded from analysis. In addition, for all outcome parameters except error proportions, error trials and the trials following an error trial were excluded to avoid the influence of post-error slowing on latencies”

Response: these lines have been moved into methods sections of both experiments respectively.

5. “If I understood correctly, the measure T-PWS was not valid for mats 3 and 4. Is this correct? If yes, this no longer seems like a useful dependent measure given that the large number of trials that were discarded.”

Response: all mentioning of force plate recordings and derived preparatory weight shift outcome parameters has been removed from the manuscript.

6. “For experiment 2, it would be advantageous to indicate whether the sequence specific learning effects (middle columns; Table 3) are significant WITHIN each age group. The last column collapses across the two groups and shows significant learning effects, but this appears to benefit from the large n (collapsed across young and old) and effects that are disproportionately larger in young (although not sig. different from older). In brief, it does not appear that the older adults exhibit significant learning per se on many metrics. This information would be of interest. Note that this does not take away from the main between-group comparisons, but would be worth specifying nonetheless.”

Response: separate t-test for within-group significance of learning effects have been calculated and added to Table 3.

7. “I leave this up to the authors but I am not convinced the methods of the two experiments cannot be presented within a single methods section followed by results of the two experiments.”

Response: in an earlier draft of the manuscript before submission the methods sections in Experiments 1 and 2 were unified. However, the authors decided to split up the methods sections and prefer to keep this separation.

Additional requirements

8. “We note that the grant information you provided in the ‘Funding Information’ and ‘Financial Disclosure’ sections do not match. When you resubmit, please ensure that you provide the correct grant numbers for the awards you received for your study in the ‘Funding Information’ section.”

Response: the grant information in funding information and financial disclosure statements have been matched.

9. “We note that you have provided funding information that is not currently declared in your Funding Statement. However, funding information should not appear in the Acknowledgments section or other areas of your manuscript. We will only publish funding information present in the Funding Statement section of the online submission form.”

Response: the funding information has been removed from the acknowledgements section.

Reviewer 1 comments

10. “The locomotor SRTT has three different step types (forward step, backward step, foot on same spot), two of which (forward, backward) could have followed a step by the same foot, or by the opposite foot. It seems that these five conditions (forward/same foot, forward/different foot, backward/same foot, backward/different foot, same spot) present different postural and balance concerns and difficulty, which could result in different challenges to older adults. The finger movement SRTT literature demonstrates aging effects increase for higher-order sequence information and other forms of increased complexity (e.g., Howard et al., 2004, Psychology and Aging; Bennett et al., 2007, Journals of Gerontology - Series B); it may be then that in a locomotor SRTT, aging effects become apparent for the more difficult/complex movements. Did step type have any impact on sequence learning, or interact with age effects on sequence learning?”

Response: we preformed this suggested analysis and included it in the methods, results and discussion sections of Experiment 2 and the general discussion. Although, stepping complexity effects were found, we observed no interactions with age group.

11. “Line 105: The quick mention of generalization feels out of place in the paragraph on chunking; I would recommend separating these concepts, and perhaps expanding on the generalization point.”

Response: the paragraphs referring to chunking, generalization and representations have been removed from the introduction.

12. “Line 257: What is meant by “Each participant received an individualized set of three randomized sequences”? I presume that this means the sequences used had the same archetypal format across participants (i.e., A-C-D-B-A-C), but that the mapping of [1,2,3,4] to [A,B,C,D] is what was randomized three times for each participant. If this is instead referring to a loop within the random condition, that should be clarified, and potential predictability driven by that loop would need to be addressed in all analyses. Could you please clarify this point?”

Response: this statement has been rephrased to express that each participant received completely randomized blocks 1, 2, and 9. At the start of the experiment the random sequence of elements were generated anew for each participant.

13. “Line 309-310: This sentence is unclear; I believe it is missing a phrase (“In addition, for response latencies but not ???, error trials and…”)”

Response: this sentence has been rephrased to clarify its meaning.

14. “Table 1: It would be good to include the random condition means (Block 9) in this Table for direct comparison with the sequence condition means. Also, assumedly EP means error proportion, but that is not specified- and if the units are % it cannot be a proportion.”

Response: the random condition means move been included in the Tables 1 and 3. Also, “error proportions” have been rephrased as “percent error” in the entire manuscript.

15. “Line 381: It is good that participants with severe hearing impairments were excluded, but what about non-severe aging-related hearing loss? Given that the cues were auditory, even moderate presbycusis could cause additional cue processing time for older adults that would affect responses in a manner unrelated to movement planning. Were audiometric thresholds obtained for older adults, or was hearing assessed in any other objective manner?”

Response: the description of the recruitment procedure for the older adults in Experiment 2 

has been reformulated. Although participants were contacted via a hospital database, any audiometric thresholds were not assessed. Nevertheless, all participants were able to understand the auditory target cues acoustically.

16. “Figure 3: What do the red and blue traces in the CoP position time series plots differentiate? I am assuming ML and AP, but it should be labelled. Also, perhaps this is just an aspect of the coordinate system used for measurements, but if this Figure illustrates a forward step by the left foot, why are left foot displacement and velocity negative?”

Response: the figure has been edited to that a forward foot displacement results in a flipped, positively increasing trajectory. The original figure showed the raw data in the kinematic coordinate frame, which had negative in the forward stepping direction.

17. “Line 471: Though the Chi-squared test did not reach significance, the descriptive statistics suggest an aging effect here. Moreover, for older adults the full range of number-of-elements-reported is given, whereas only the lower bound is mentioned for young adults. Did older and young adults differ on number of sequence elements correctly reported?”

Response: the average number of elements in a correct sequence was calculated and tested between both age groups. No differences were found. This was included in the results section.

18. “Line 496: After stating the lack of age x sequence condition interaction, it would be helpful to report the lower bound of the observed p-value (e.g., p > 0.2). This comment applies to all such statements of null effects intended to be meaningful in the Results sections.”

Response: for all observed non-significant p-values, the F or t statistics, p values and partial eta-squared have been included in the manuscript.

19. “Line 533: What was the reasoning for considering the T-PWS, T-STEP, and T-TARGET measures to be the main movement parameters? If these are the parameters of primary interest, why were multiple other measures included in analyses too?”

Response: an explanation was added to better explain the reasoning for the inclusion of secondary movement parameters.

20. “Line 550: The comma between “participants” and “who” disrupts the meaning of the sentence. There are multiple odd commas like this throughout the manuscript- these should be identified and removed.”

Response: we have probably overgeneralized german punctuation rules concerning relative clauses and inserted superfluous commas. We tried to identify and remove those.

21. “Line 553: Would be good to remind here that the reason T-STEP and T-PWS are not mentioned is that those measures were not possible in Experiment 1. You mention this later (line 580), but it would be useful here.”

Response: a reminder has been added that the T-STEP parameter etc. were not available in Experiment 1.

22. “Line 562: I don’t recall seeing the T-PWS sequence effect being reported as marginal in the Results. I see in Table 3 the p-value for that comparison was 0.17, which is a stretch for including as marginal- the authors do hint here though that the effect was more reliable in young adults alone, which is not reported. Given the lack of age x sequence interaction though, such a breakdown would not be a licensed comparison. This claim should either be better defended, or removed.”

Response: this statement has been removed.

23. “Line 601: The given interpretation that locomotor sequence learning may be less susceptible to aging decline than finger movement sequence learning is certainly reasonable. An alternate interpretation the authors may want to consider is that the aging effects reported in the finger movement literature may instead reflect a relative expertise of young adults for finger sequence movements, rather than an aging-related decline, and that this aging-related difference in expertise is not observed for stepping movements.”

Response: a statement has been added to the discussion which considers a possible specialization of younger adults in finger activities with respect to the interpretation of aging effects in the literature.

Reviewer 2 comments

24. “It is unclear whether accuracy was emphasized to participants. What instructions were participants provided about the speed and accuracy of their movements? This may affect how quickly and accurately they respond to the auditory stimuli.”

Response: in order to clarify the instructions regarding emphasis on speed or accuracy we moved this information to a more prominent position.

25. “Was a recall or recognition test used to assess whether participants were able to determine that there was a sequence in the stimuli? It seems so, as some results are described, but the procedure and questions asked should be described in the methods.”

Response: the description of the procedure for the recall and recognition has been rephrased with additional detail.

26. “What were the results of the paired t-tests in Experiment 1 and the ANOVA in Experiment 2? The methods describe these analyses, but they are not reported in the results.”

Response: we reformulated sentences in the methods and results sections to signpost the results of paired t-tests and ANOVA better in both experiments.

27. “What is the purpose of the Pearson correlations? What part of the research question were they trying to answer?”

Response: we added further explanations of the purpose of the Pearson correlations.

28. “As older adults may make more anticipatory postural adjustments, how were they accounted for with respect to the postural adjustments all participants made to get ready to take a step? Was there any difference in these augments between the older and young adults?”

Response: as all force plate measures have been removed from the manuscript a difference between old adults and young regarding APAs is no longer reasonable in the context of the experiments. It is an interesting question indeed but would need a follow-up study (currently in progress).

29. “The authors statement that, “Surprisingly, despite decreased functional reserves and reduced locomotor stability that occurs with aging (i.e. chronological age), it had no obvious influence on the acquisition of the stepping sequence in terms of any latencies, movement parameters or error rates” (lines 594-596) is a bit confusion. It seems from Figure 4 that older adults were overall slower across all blocks and made more errors. Perhaps the ANOVA results would clarify this.”

Response: in order to improve the clarity of this statement this section has been edited slightly.

30. “The conclusion seems a bit overstated and broad. For example, how can this task be used as a clinical tool? Without assessing the task on a clinical population, it is unclear which parameters would be valuable for assessing a clinical population, especially since the authors did not find any significant differences between the older and young adults.”

Response: a paragraph has been added to the general discussion section to suggest possible clinical applications of the paradigm.

31. “This is a minor stylistic suggestion. There are many abbreviations used in the manuscript that disrupt the flow of the paper. Some abbreviations are defined, but never referred to again (e.g., AR in line 157) that can be taken out.”

Response: we reduced the number of abbreviations.

---

## [Decision Letter · Decision Letter 1]

17 Mar 2022

PONE-D-21-22593R1Keeping in step with the young: chronometric and kinematic data show intact procedural locomotor sequence learning in older adultsPLOS ONE

Dear Dr. Johannsen,

Thank you for submitting your manuscript to PLOS ONE. After careful consideration, we feel that it has merit but does not fully meet PLOS ONE’s publication criteria as it currently stands. Therefore, we invite you to submit a revised version of the manuscript that addresses the points raised during the review process.

We look forward to receiving your revised manuscript.

Kind regards,

Bradley R. King

Academic Editor

PLOS ONE

Journal Requirements:

Reviewers' comments:

Reviewer's Responses to Questions

**Comments to the Author**

1. If the authors have adequately addressed your comments raised in a previous round of review and you feel that this manuscript is now acceptable for publication, you may indicate that here to bypass the “Comments to the Author” section, enter your conflict of interest statement in the “Confidential to Editor” section, and submit your "Accept" recommendation.

Reviewer #1: All comments have been addressed

Reviewer #2: All comments have been addressed

2. Is the manuscript technically sound, and do the data support the conclusions?

Reviewer #1: Yes

Reviewer #2: Yes

3. Has the statistical analysis been performed appropriately and rigorously? 

Reviewer #1: No

Reviewer #2: Yes

4. Have the authors made all data underlying the findings in their manuscript fully available?

Reviewer #1: Yes

Reviewer #2: Yes

5. Is the manuscript presented in an intelligible fashion and written in standard English?

Reviewer #1: Yes

Reviewer #2: Yes

6. Review Comments to the Author

Reviewer #1: Keeping in step with the young: chronometric and kinematic data show intact procedural locomotor sequence learning in older adults

The authors’ additions and edits have largely addressed the concerns I expressed in my initial review of this manuscript, and it is informative to see that although step complexity did affect step characteristics, this did not interact with age. I do believe performing a similar analysis on the data from Experiment 1 would also be informative, giving a sense of the reliability of the complexity effects.

My remaining suggestions for improving the manuscript are as follows:

1) The analysis breakdown by step complexity for Experiment 2 is welcome, and fully addresses my concern that more complex movements might reveal aging-related impacts on sequence learning. The non-age-dependent effects of step complexity on kinematic measures reported for Experiment 2 are also informative for future work that may use similar designs.

However, it is unclear why the complexity analyses were only performed for Experiment 2; there is no obvious reason why similar analyses could not have also been performed on Experiment 1, which would give a sense of reliability to the complexity effects. This would also make sense conceptually, as Experiment 1 is introduced as a way to validate the approach used in Experiment 2, so it is unclear why an unvalidated additional analysis is introduced for Experiment 2.

2) A power analysis based on data from Experiment 1 indicated that n = 9 was sufficient for detecting large within-group response latency effects, and previous work used ns between 15–40 to demonstrate age effects. However, these points are then used to justify a sample size of 22 as an a priori target for detecting large age effects across a number of dependent variables; it is not clear how those two data points support that conclusion. It seems that 22 was the minimum group n, and the power justification is more post-hoc than a priori. Wouldn’t it make more sense then to do a post-hoc power calculation instead and state what effect sizes you had power to detect?

3) For Experiment 2, why were ANOVAs used only for T-STEP and T-TARGET? Mixed ANOVA seems an appropriate statistical choice for all dependent variables analyzed here.

4) Figure 1: The caption here suggests that the reflective markers for the optoelectronic system were worn for both Experiment 1 and Experiment 2, but if I understand correctly the optoelectronic system was only employed for Experiment 2. If that is correct this caption needs to reflect that.

5) Line 352, what is meant by “I presentation of the auditory stimuli”?

6) Line 371, “was defined” seems extraneous.

7) Line 382, “mediolateral” is not in Figure 3 and doesn’t need an acronym defined here.

8) Lines 384 – 408, this first paragraph in the Experiment 2 Statistical Analysis section would benefit from being broken into 2-3 smaller paragraphs.

9) Lines 476 – 477, it is claimed that Table 3 contains the t-test and ANOVA results for all DVs, but Table 3 only contains t-test results.

10) Table 3: STEP-DUR t-test results for each group are not significant, but bolded.

11) Table 4: Bolding is inconsistent in Table 4 too. Unless alpha in Table 4 was 0.001? If so, that should be specified somewhere.

12) Table 5: What do the italics indicate? Also, please define acronyms in Table caption.

13) Lines 600 – 606, Although the group differences in explicit knowledge did not reach significance, the group means were consistent with the extant literature. So, I would not lean too hard on the current null finding as counter-evidence to prior demonstrations of age differences in explicit sequence knowledge during the SRTT.

Reviewer #2: (No Response)

7. PLOS authors have the option to publish the peer review history of their article (what does this mean?). If published, this will include your full peer review and any attached files.

Reviewer #1: No

Reviewer #2: No

---

## [Author Response · Author response to Decision Letter 1]

24 Mar 2022

Dear Dr. King and Reviewers,

once again, we like to express our gratitude for editing and reviewing our submitted manuscript. We have taken the additional feedback and suggestions seriously and addressed them in the manuscript accordingly. In the following overview, we have listed each recommendation and our response and the changes made to the manuscript.

Reviewer 1 comments

1) The analysis breakdown by step complexity for Experiment 2 is welcome, and fully addresses my concern that more complex movements might reveal aging-related impacts on sequence learning. The non-age-dependent effects of step complexity on kinematic measures reported for Experiment 2 are also informative for future work that may use similar designs. However, it is unclear why the complexity analyses were only performed for Experiment 2; there is no obvious reason why similar analyses could not have also been performed on Experiment 1, which would give a sense of reliability to the complexity effects. This would also make sense conceptually, as Experiment 1 is introduced as a way to validate the approach used in Experiment 2, so it is unclear why an unvalidated additional analysis is introduced for Experiment 2.

Response: we have now performed and included the analysis of the effect of step complexity for Experiment 1. This change required alterations to the methods sections of both Experiments for the sake of consistency. Similarities between both experiments are discussed in the summary section of Experiment 2.

2) A power analysis based on data from Experiment 1 indicated that n = 9 was sufficient for detecting large within-group response latency effects, and previous work used ns between 15–40 to demonstrate age effects. However, these points are then used to justify a sample size of 22 as an a priori target for detecting large age effects across a number of dependent variables; it is not clear how those two data points support that conclusion. It seems that 22 was the minimum group n, and the power justification is more post-hoc than a priori. Wouldn’t it make more sense then to do a post-hoc power calculation instead and state what effect sizes you had power to detect?

Response: we have followed the reviewer’s suggestion and modified the paragraphs in the methods section of Experiment 2 in terms of a discussion of a post-hoc power analysis. In the General discussion section we are also now referring to the power issue. 

3) For Experiment 2, why were ANOVAs used only for T-STEP and T-TARGET? Mixed ANOVA seems an appropriate statistical choice for all dependent variables analyzed here.

Response: thank you for pointing this out. This statement is clearly mistaken, it represented an earlier state of the draft, and has been corrected. Of course, mixed ANOVAs were also applied for the remaining dependent variables.

4) Figure 1: The caption here suggests that the reflective markers for the optoelectronic system were worn for both Experiment 1 and Experiment 2, but if I understand correctly the optoelectronic system was only employed for Experiment 2. If that is correct this caption needs to reflect that.

Response: we have extended the caption of this figure to state that the markers were only used for Experiment 2.

5) Line 352, what is meant by “I presentation of the auditory stimuli”?

Response: this was a typo and has been corrected.

6) Line 371, “was defined” seems extraneous.

Response: has been removed.

7) Line 382, “mediolateral” is not in Figure 3 and doesn’t need an acronym defined here.

Response: has been removed.

8) Lines 384 – 408, this first paragraph in the Experiment 2 Statistical Analysis section would benefit from being broken into 2-3 smaller paragraphs.

Response: this section has been broken down into smaller paragraphs.

9) Lines 476 – 477, it is claimed that Table 3 contains the t-test and ANOVA results for all DVs, but Table 3 only contains t-test results.

Response: has been corrected by removing the reference to any ANOVA results

10) Table 3: STEP-DUR t-test results for each group are not significant, but bolded.

Response: has been unbolded.

11) Table 4: Bolding is inconsistent in Table 4 too. Unless alpha in Table 4 was 0.001? If so, that should be specified somewhere.

Response: we have stated in the statistics section now, that a more conservative significance criterion was chosen.

12) Table 5: What do the italics indicate? Also, please define acronyms in Table caption.

Response: the italics only served a purpose during preparation of the manuscript and have been removed. Also, we explained the acronyms in the table.

13) Lines 600 – 606, Although the group differences in explicit knowledge did not reach

significance, the group means were consistent with the extant literature. So, I would not lean too hard on the current null finding as counter-evidence to prior demonstrations of age differences in explicit sequence knowledge during the SRTT.

Response: we have softened our conclusion accordingly and addressed the issue of statistical power here.

---

## [Editor Report · Decision Letter 2]

28 Mar 2022

Keeping in step with the young: chronometric and kinematic data show intact procedural locomotor sequence learning in older adults

PONE-D-21-22593R2

Dear Dr. Johannsen,

We’re pleased to inform you that your manuscript has been judged scientifically suitable for publication and will be formally accepted for publication once it meets all outstanding technical requirements.

Kind regards,

Bradley R. King

Academic Editor

PLOS ONE
---

## [Editor Report · Acceptance letter]

25 Apr 2022

PONE-D-21-22593R2 

Keeping in step with the young: chronometric and kinematic data show intact procedural locomotor sequence learning in older adults 

Dear Dr. Johannsen:

I'm pleased to inform you that your manuscript has been deemed suitable for publication in PLOS ONE. Congratulations! Your manuscript is now with our production department. 

Kind regards, 

on behalf of

Dr. Bradley R. King 

Academic Editor

PLOS ONE